# The Associations between Family-Related Factors and Excessive Internet Use in Adolescents

**DOI:** 10.3390/ijerph17051754

**Published:** 2020-03-08

**Authors:** Anna Faltýnková, Lukas Blinka, Anna Ševčíková, Daniela Husarova

**Affiliations:** 1Faculty of Social Studies, Masaryk University, Brno 601 77, Czech Republic; anafaltynkova@gmail.com (A.F.); asevciko@fss.muni.cz (A.Š.); 2Department of Health Psychology and Methodology Research, Faculty of Medicine, P.J. Safarik University in Kosice, Košice 040 01, Slovakia; daniela.husarova@upjs.sk

**Keywords:** adolescent internet use, excessive internet use, internet addiction, family factors, parenting styles

## Abstract

This study examined the relationship between Excessive Internet Use (EIU) in adolescents and their family environment, namely the family type, the family economic status, the effect of parental care, the level of parental control, the amount of parental monitoring, the quality of communication, and the time spent together. The study was based on data from an international survey, Health Behaviour in School Aged Children (HBSC), conducted in Slovakia. The sample representative for adolescents included 2547 participants (51% boys) aged 13–15. Multiple-step linear regression revealed that higher parental care and parental monitoring predicted lower EIU, while higher parental overprotection and lower socioeconomic status predicted higher EIU. The results suggest that both so-called optimal parenting (i.e., the balance of emotional warmth and protection) and the adolescent′s autonomy lower the risk of EIU. Family factors explained about 14% of the variance, which suggests that aside from personal, cognitive and affective factors, a close social environment also plays an important role in adolescence EIU.

## 1. Introduction

The internet and the use of various digital devices are among the most important socialization factors and leisure activities in the lives of adolescents. The extent and intensity of internet usage has spawned research on both the benefits and the risks of online activities, including their addictive potential. This facilitated the inclusion of Internet Gaming Disorder in the International Classification of Diseases 11th Revision (ICD-11, World Health Organization). Generalized internet addiction has not reached such a consensus in the scientific community and various terms have been proposed (e.g. internet addiction, compulsive internet use [1]; pathological internet use [2]; problematic internet use [3]; internet use disorder [4]. In this paper, we work with the term Excessive Internet Use (EIU) and define it as a problem that manifests as preoccupation, mood changes, difficulties with limiting time online, and subsequent conflicts, including difficulties that belong among the symptoms of addictive behaviour. In our study, EIU represents an umbrella term for any excessive online behaviour. Although some older literature suggests the existence of generalised internet addictions [2], more current literature leans towards specific internet use disorders [4]. This has been supported by empirical findings that generalised and specific internet addiction mostly overlap [5], and that some online applications, especially online gaming [6] and social networking [7], contribute to this phenomenon more than others.

Adolescents are often shown to be the most vulnerable group in terms of developing EIU [8]. Their vulnerability stems from specific immaturity that is characteristic for the developmental period and includes increased impulsivity, impaired self-regulation, and higher tendencies toward reward-seeking behaviours [9,10]. Given the fact that the internet is a readily available tool that is thoroughly embedded in their everyday routines [11], EIU may become an issue for some adolescents. Higher levels of internet use have been linked to a range of problems, such as nervousness and irritability due to lack of sleep [12], increased depression [13], health problems caused by sedentary behaviour [14,15], and neglecting academic and other responsibilities [16,17].

Prior research has mostly investigated various individual risk factors that range from low self-esteem and self-efficacy [18,19], self-directedness [20], neuroticism [21], and loneliness [22] to attention difficulties [23]. As such, the role of social environment is neglected in models that describing both generalised and specific disordered internet use-only the role of social cognition (e.g., feelings of isolation, perceived lack of social support) is mentioned in the Interaction of Person-Affect-Cognition-Execution (I-PACE) model [4] and the Cognitive-Behavioral Model of Pathological Internet Use [2]. However, theories of problematic adolescent behaviour, e.g., the Problem Behavior Theory [24] and theories for media effects, e.g., the Differential Susceptibility to Media Effects Model [25] propose stronger consideration of the environmental influence, notably the family, which may both predictand moderate adolescent behaviour, including patterns of internet use.

The family environment is the closest social environment for adolescents and it plays an essential role in development [26]. Previous research suggests that the presence of a dysfunctional and high-conflict family environment may increase the likelihood of developing various forms of adolescent pathological behaviour, like alcohol use and gambling [27,28], as well as developing excessive online behaviour [29]. Specifically, dysfunctional parent–child communication [30], the existence of family conflict [31], and the lower emotional availability of parents [32] have been found to be associated with EIU in adolescents. More parental emotional warmth and care have been shown to be protective factors against EIU [32,33].

Parents face challenges as their adolescent grows. Adolescents develop their own individuality and learn to take responsibility for their own actions. Their needs for privacy and independence gain importance [34], but at the same time compete with needs for closeness and relatedness, which still remains significant [35]. Insufficient social support in this developmental stage can intensify feelings of loneliness or social isolation, which often appear in adolescence [36]. Indeed, the levels of social support have been found to be associated with EIU [37]. Therefore, parents have to find a balance between promoting autonomy and monitoring behaviour, which also applies to internet use. According to existing literature, restriction and control are associated with negative outcomes, while parental monitoring promotes healthy and positive psychological development [38]. With respect to internet use, the adequate monitoring of an adolescent´s whereabouts and the strategies of behavioural control have been linked to lower levels of EIU [39,40], while intrusive parental control and autonomy restriction are likely to manifest as risk factors for EIU [41].

The present overview suggests that family environmental factors might play an important role in moderating internet use in adolescence. However, to our knowledge, none of the available studies have comprehensively examined these factors in relation to EIU. Therefore, the aim of this study is to analyse the associations between EIU and an array of family-related factors—family composition, socioeconomic status, parenting strategies (e.g., care, monitoring, overprotection), quality of communication, and time spent together, while controlling for adolescent gender, age, and digital screen time. We assume that adolescents, who experience sufficient social support, care, and personal autonomy will report lower scores for EIU than those who perceive their parents as unloving, too controlling, and restrictive of their personal freedom. 

## 2. Materials and Methods 

### 2.1. Data Collection

We used data from the Health Behaviour in School-aged Children (HBSC) study, which was a cross-national survey conducted in Slovakia in 2014. A two-step procedure was used for data collection in order to obtain a nationally representative sample of adolescents. In the first step, 151 elementary schools were randomly selected from the register of all eligible schools in Slovakia, as provided by the Slovak Institute of Information and Prognosis for Education. In the second step, every school that agreed to participate had one randomly selected class per grade within the target group of students (i.e., fifth to ninth grades). Standardized self-reporting questionnaires were administered by trained administrators and completed by individual students during a classroom session. The study was approved by the Ethics Committee of the Medical Faculty at P.J. Safarik University in Kosice (No: 9/2012). All participation was fully voluntary and anonymous. Parental consent was obtained before administration.

### 2.2. Sample

A total of 10,179 Slovak adolescents aged 11–15 participated in the survey and completed the self-report questionnaire. Measures listed below were offered only to these children. Items to measure excessive internet use were included only in version A of the questionnaire, which was designed for students aged 13–15 (only 3135 adolescents were given version A). We also eliminated participants who did not provide 50% of the values for any of the variables of interest. The final sample consisted of 2547 adolescents from 13 to 15 years old (*M* = 14.32, *SD* = 0.91), with 50.9% being boys.

### 2.3. Measures

HBSC followed the strict methodology of the survey, which also applied to the translation process. All of the measures used in the study were translated from English to Slovak using the back-translation procedure. Moreover, back-translations are also independently checked by a translation team within the HBSC network, which was specifically established for this task.

EIU was measured using the *Excessive Internet Use* scale (EU Kids Online network; eukidsonline.net). This scale was validated in 25 European countries [42]. It consists of five items that cover five of the six factors of the components for the model of behavioural addiction [43]: salience (“I have gone without eating and sleeping because of the internet“); withdrawal symptoms (“I have felt bothered when I cannot be on the internet“); tolerance (“I have caught myself surfing when I am not really interested“); relapse (“I have tried unsuccessfully to spend less time on the internet“); and conflict (“I have spent less time than I should with either family, friends, or doing schoolwork because of the time I spend on the internet“). Using a four-point scale (ranging from 1 = “never“ to 4 = “very often“), participants rated how often they had experienced the symptoms in the preceding12 months. The final variable was created as a mean of the five items, with higher numbers representing more excessive internet use. The scale was reliable. Cronbach′s alpha equaled 0.79.

*Digital screen time* was assessed as the total time spent on the computer during an average weekday. The scale had nine options that ranged from zero hours a day to seven and more hours a day. Digital screen time was previously associated with the greater duration of internet use per week and with internet addiction [44].

*Socioeconomic status* was assessed by a question about the perceived financial situation of the family (1 = “our family is not at all well off” to 5 = “our family is very well off”).

*Family composition* was constructed as a dichotomous variable, with 0 representing an incomplete family (e.g., the child lives with only one parent and/or step-parent, or with none of the parents) and 1 representing a complete family (e.g., the child lives with both parents in one household).

*Parental care* was assessed with a composite score that combines information from two scales that measure identical construct. Eight items were taken from the Parental Bonding Instrument: A Brief Current Form (PBI-BC [45]) and four items came from the Multidimensional Scale of Perceived Social Support (MSPSS [46]). Items in both subscales measure parental warmth, interest, and supportive behaviour from the adolescents′ perspective (e.g., “My mother/my father helps me as much as she/he can”; “My family is willing to help me make decisions.”). The scale had a single-factor structure and the composite score for the variable was constructed as a mean score of 12items for the mother and father. For children who had only one parent, the mean score was computed for one parent. This allowed us to avoid reducing the statistical power because of the exclusion of children from single-parent families. The scale was reliable and Cronbach′s alpha equaled 0.88.

*Parental overprotection* was measured with eight items from the Parental Bonding Instrument: A Brief Current Form (PBI-BC [45]). The scale is designed to measure parental control and overprotective behaviour from the perspective of the adolescent (e.g., “Mother/father likes me to make my own decisions“). Participants completed scales for mothers and fathers separately, and each item was rated on a three-point scale according to the perceived frequency of certain behaviour (1 = “never“; 2 = “sometimes“; 3 = “almost always“). The variable Parental Overprotection was constructed as a mean score of items for paternal overprotection and maternal overprotection. For children who had only one parent, the mean score was computed for one parent. The scale was reliable and Cronbach′s alpha equaled 0.65.

*Parental monitoring* was measured with a five-item scale for how much the parents know about the following aspects of their adolescent′s life: how they spend their free time; what they do after school; where they go in the evening; how they spend their money; and who their friends are. Response options were 1 = “she/he knows a lot“; 2 = “she/he knows a little“; and 3 = “she/he knows nothing“. The mean score was computed from all 10items for the mother and father. For children who had only one parent, the mean score was computed for one parent. The scale was reliable and Cronbach′s alpha equaled 0.88.

*Time spent together* was measured by eight items that assessed how often adolescents engaged in certain common activities with their parents, like playing indoor games, visiting relatives, and watching television. For each item, participants were asked to choose the most appropriate of the following five response categories: 1 = “never”; 2 = less often”; 3 = “about once a week”; 4 = “most days”; and 5 = “every day”. The variable was constructed as a mean score of all eight items. The scale was reliable and Cronbach′s alpha equaled 0.85.

*Quality of communication with parents* was measured by asking respondents how easy was for them to talk to their parents about things that bothered them (1 = “very easy“ to 4 = “very difficult“).The mean score was computed from both items for the mother and father. For children with only one parent we used the single score they chose for that parent. The scale was reliable and Cronbach′s alpha equaled 0.68.

Descriptive statistics and Cronbach′s alpha values for all of the scales are presented in Table 1.

### 2.4. Data Analysis

The data were analysed using IBM SPSS Statistics 25 software. Spearman´s correlation coefficients were calculated for all of the variables. Gender differences in the level of excessive internet use and other variables were assessed using the Mann–Whitney U-test. Nonparametric tests were used because the majority of variables were not normally distributed (values of skewness and kurtosis are presented in Table 1). To determine the associations between excessive internet use and family factors, we conducted a hierarchical multiple regression that controlled for the gender and age of the participants and for the frequency of computer use. Variables were entered into the regression in two steps with the Enter method. In the case of sufficiently large sample sizes, regression analysis provided valid results, even for non-normally distributed variables [47]. Control variables were entered in the regression in the first step. All of the family factors were entered in the second step. The assumptions of regression analysis were met: the VIF score for any variable was below 1.94 (tolerance above 0.52), indicating no problem with multicollinearity, and the residuals were uncorrelated (Durbin–Watson coefficient = 1.88) and approximately normally distributed.

## 3. Results

The means, standard deviations, and Spearman´s rank correlation coefficients of the studied variables are presented in Table 2. On average, participants reported relatively low scores on excessive internet use (*M* = 1.62, *SD* = 0.61). The majority of the sample obtained the minimum possible score in the EIU scale; thus, the distribution of the dependent variable was positively skewed. Excessive internet use was positively related to adolescent age, frequency of computer use, and parental overprotection, and negatively related to all other family variables.

A Mann–Whitney U test was conducted to compare the level of excessive internet use in boys (*M* = 1.62, *SD* = 0.64) and girls (*M* = 1.62, *SD* = 0.59). There was no significant difference (*U* = 806670.5, *p* > 0.05, *η^2^* = 0.01). There were also no differences between boys and girls in parental care (*U* = 807552.5, *p* > 0.05, *η^2^* = 0.01), parental overprotection (*U* = 783925.0, *p* > 0.05, *η^2^* = 0.06), parental monitoring (*U* = 794794.0, *p* > 0.05, *η^2^* = 0.03), communication (*U* = 806933.0, *p* > 0.05, *η^2^* = 0.01), time spent together (*U* = 809466.5, *p* > 0.05, *η^2^* = 0.002), socioeconomic status (*U* = 800034.5, *p* > 0.05, *η^2^* = 0.02), family composition (*U* = 791531.0, *p* > 0.05, *η2* = 0.04), and digital screen time (*U* = 795816.5, *p* > 0.05, *η^2^* = 0.03).

Regression coefficients for each predictor are presented in Table 3.

The first model, including age, gender, and frequency of computer use, significantly predicted excessive internet use (*F*(3, 2544) = 49.14; *p* < 0.001) and accounted for 5.5% of the variance of the dependent variable. Both age (*t*(2544) = 3.69; *p* < 0.001) and digital screen time (*t*(2544) = 11.09; *p* < 0.001) significantly predicted EIU.

After entering family-related variables in the second step, the proportion of explained variance increased to 14% and this model also significantly predicted EIU (*F*(10, 2537) = 41.37; *p* < 0.001). Our analysis revealed that higher socioeconomic status (*t*(2537) = −2.44; *p* < 0.05), parental care (*t*(2537) = −3.19; *p* < 0.01), and parental monitoring (*t*(2537) = -6.62; *p* < 0.001) were significant negative predictors (i.e., protective factors) of EIU. Parental overprotection (*t*(2537) = 8.14; *p* < 0.001) and time spent together (*t*(2537) = 3.75; *p* < 0.001) significantly predicted EIU in the opposite direction, indicating that those variables figure as risk factors. The associations between family composition (*t*(2537) = 0.53; *p* > 0.05), parental communication (*t*(2537) = 0.23; *p* > 0.05), and excessive internet use were not significant. Age and frequency of computer use remained significant predictors, whereas association between gender and excessive internet use remained insignificant.

## 4. Discussion

This study examined family environmental factors in relation to adolescent Excessive Internet Use. The findings show that the strongest protective factor was the parental monitoring of the adolescent′s activities (e.g., knowing about friends, knowing how they spend free time), followed by parental care (e.g., emotional warmth, overall support within the family environment). On the other hand, the strongest risk factor associated with a higher score of EIU was parental overprotection (e.g., parental behaviour that decreases adolescents′ independence), followed by increased time spent together and lower socioeconomic status of the family. 

These findings point to two important issues. First, our study supports the so-called optimal parenting that seems to be an ideal parenting strategy that is associated with lower scores for EIU [48]. Parents who use this strategy might be successful at balancing their care for their adolescents and their protection, while, at the same time, respecting their personal autonomy [49]. As the need for autonomy in adolescence increases with age, autonomy restrictions may lead to diverse problems, including lower self-esteem or self-efficacy. Sufficient support for autonomy has a protective function, which has also been shown in previous studies [50].

Second, from the perspective of associations among family environmental factors and problematic behaviour in youth, our results show a mix of similarities and dissimilarities between EIU and other traditional problematic adolescent behaviours. In the case of traditional antisocial adolescent behaviours, such as drinking alcohol and delinquency, parental monitoring and care have generally been identified as protective factors, while parenting styles that harm the confidence and independence of children were found to be risk factors [51,52]. On the other hand, antisocial behaviours were often explained decreased family monitoring and involvement [52,53]. In this respect, EIU seems to be distinct from traditional antisocial and problematic behaviours—our results showed that overprotection and increased time at home and in the family environment (in other words, decreased time in peer groups) increase EIU. The distinction between EIU and problematic adolescent behaviour is further supported by De Looze and his colleagues [54], who found that the overall trend in the decrease of substance use in recent adolescents is not related to increased use of digital media; thus, EIU is not simply a replacement for other forms of problematic behaviour. Despite the fact that some studies found an association between EIU and various conduct problems [55,56], and that problematic adolescent behaviours and EIU share some underpinnings with ADHD [57], our study points to the need for specific parenting approaches when addressing EIU and when addressing traditional problematic behaviours. The mutual relatedness of EIU and other forms of problematic behaviours should be the subject of a more thorough investigation.

It must be noted that family factors themselves explain only about 8% (14% if we include digital screen time and age) of the variance of EIU. This is in line with previous research that showed that family factors explained 6% of the variance of EIU through parental mediation strategies [58]. It is likely that the strongest influences for EIU are personality and psychological factors, like ADHD, lower self-esteem, and emotional difficulties in general [42], which is also suggested by theoretical models like the I-PACE [4], and cognitive-behavioural models [2,59]. Since environmental factors play a non-negligible role in adolescent lives and they should be incorporated into theoretical models of EIU and specific internet disorders as has been done in general media-effect models (e.g., Differential Susceptibility to Media-Effects Model [25]). 

The strength of associations was rather weak in our study. Except for parental monitoring and overprotection, all other significant predictors were negligible. Whether they remained significant only due to the large sample size or because they indeed play some role in EIU should be the focus of further research. For example, lower social economic status, although correlated with EIU, lost most of its strength in the factor analysis. However, the literature has identified lower socioeconomic status as a risk factor of EIU [60]. In our research, lower socioeconomic status worked in the same direction as parental over protection, the time that the adolescent spent at home, and the time that the adolescent spent with digital devices. This may indicate that, in such families, there is a lack resources for meaningful structured afterschool activities (e.g., sports, music lessons) and that digital devices are used as cheap alternatives for filling free time. Such a hypothesis needs further investigation. 

The results of this study should be considered in light of several limitations. First, since the research design is cross-sectional, causal relationships between family factors and EIU could not be inferred based on our results. In this case, longitudinal studies would better examine the direction of the observed effects. Second, the data were obtained solely from adolescent self-report questionnaires. For future studies, it would be useful to complement this method of data collection with reports from parents to provide a comparison for parent–adolescent dyads. Third, the data did not include variables to measure specific internet-related mediation strategies, so we were only able to analyse general attitudes in parenting and the atmosphere in the family. In addition, digital screen time did not measure internet use, per se. Nonetheless, prior research has clearly shown that digital screen time predominantly overlaps with time spent online [61]. And, lastly, the main variable, EIU, although used in many previous studies and validated in most European countries, consists of only five items and does not allow for an in-depth investigation of the excessive use of various applications. This may be problematic from the perspective of the ongoing discussion of generalised vs. specific internet-based disorders. In this respect, our study lacked the differentiation of the applications that contribute to EIU Also, the main variable has not been validated for discriminative purposes, and it is a continuum without differentiation between healthy and pathological internets users. Despite these limitations, the main advantages of this study are the use of the national representative sample of adolescents and a several family variables that allow for their mutual control in the regression analysis. 

## 5. Conclusions

Our study revealed several important factors associated with adolescent Excessive Internet Use. The strongest protective factor was the parental monitoring of their adolescent′s activities (i.e., parents being aware of what, where, when, and with whom their adolescent children spend their leisure time). This was supported by higher care (i.e., emotional warmth, atmosphere in the family). The main risk factor of EIU appeared to be the parental overprotection (i.e., parental behaviour that harmed the respondent′s independence and confidence). Certain risk factors, although very small, were also brought by the lower socioeconomic status of the family and increased time spent at home, which were also supported by increased digital screen time. These results indicate that optimal parenting is a balance among parental care, protection, and allowing adolescents to build independence and competence. It also points to the potential role of family lifestyles associated with limited resources. 

## Figures and Tables

**Table 1 ijerph-17-01754-t001:** Descriptive statistics for study variables.

	*M*	*SD*	Skewness	Kurtosis	Cronbach´s Alpha
Excessive internet use	1.62	0.61	1.37	1.95	0.79
Age	14.32	0.91	0.18	−0.46	
Incomplete family			−1.57	0.47	
Digital screen time	4.43	2.17	0.56	−0.41	
Socioeconomic status	3.84	0.83	−0.50	0.29	
Parental care	3.73	0.66	−1.04	1.18	0.88
Parental overprotection	1.64	0.35	0.24	−0.07	0.65
Parental monitoring	2.47	0.46	−0.81	0.06	0.88
Time spent together	2.82	0.79	0.30	0.05	0.85
Communication	2.81	0.81	−0.34	−0.44	0.68

**Table 2 ijerph-17-01754-t002:** Spearman´s rank correlations of all study variables.

	1	2	3	4	5	6	7	8	9	10
1. Excessive internet use	-									
2. Age	0.10 **	-								
3. Digital screen time	0.23 **	0.12 **	-							
4. Socioeconomic status	−0.13 **	−0.08 **	−0.02	-						
5. Family composition	−0.03	−0.01	−0.05 *	0.14 **	-					
6. Parental care	−0.24 **	−0.10 **	−0.05 *	0.27 **	0.03	-				
7. Parental overprotection	0.23 **	−0.03	0.05 **	−0.12 **	−0.07 **	−0.34 **	-			
8. Parental monitoring	−0.28 **	−0.10 **	−0.17 **	0.17 **	0.14 **	0.52 **	−0.28 **	-		
9. Time spent together	−0.13 **	−0.16 **	−0.08 **	0.26 **	0.06 **	0.46 **	−0.11 **	0.37 **	-	
10. Parental communication	−0.17 **	−0.09 **	−0.03	0.23**	−0.01	0.51 **	−0.22 **	0.31 **	0.33 ^**^	-

^*^*p* < 0.05, ^**^*p* < 0.01.

**Table 3 ijerph-17-01754-t003:** Regression coefficients.

	Model 1			Model 2		
	*B*	*SE*	*β*	*B*	*SE*	*β*
Constant	0.65	0.19		1.01	0.23	
Age	0.05	0.01	0.07 ***	0.04	0.01	0.06 ***
Gender	0.01	0.02	0.01	0.02	0.02	0.02
Digital screen time	0.06	0.01	0.22 ***	0.05	0.01	0.18 ***
Socioeconomic status				−0.04	0.02	−0.05 *
Family composition				0.02	0.03	0.01
Parental care				−0.08	0.02	−0.08 ***
Parental overprotection				0.29	0.04	0.17 ***
Parental monitoring				−0.20	0.03	−0.15 ***
Time spent together				0.06	0.02	0.08 ***
Communication				0.01	0.02	0.01
R	0.23			0.37		
R^2^	0.06			0.14		
Adjusted R^2^	0.05			0.14		

* *p* < 0.05, *** *p* < 0.001.

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
