# Peer review of "The Associations between Family-Related Factors and Excessive Internet Use in Adolescents"

_ijerph, 2020, doi:10.3390/ijerph17051754_

Round 1

Reviewer 1 Report

The authors take into account the time that participants spend
in front of the computer screen. It would have been interesting to see
what part of that time they spend on the internet. It is possible that
a part of the time that the participants invest in the computer
is destined to the realization of works, to the use of programs
related to their academic activities.

The authors talk about the time students spend on the Internet from the time they are in front of the computer screen. This measure may be incorrect, as it is possible that students are in front of the computer screen without an internet connection. It is possible that being in front of the computer screen means that they are doing some academic activity without an internet connection. Please explain this.

Author Response

The authors talk about the time students spend on the Internet from the time they are in front of the computer screen. This measure may be incorrect, as it is possible that students are in front of the computer screen without an internet connection. It is possible that being in front of the computer screen means that they are doing some academic activity without an internet connection. Please explain this.

RESPONSE:

We thank the reviewer for their positive feedback and also for the main comment. Unfortunately, we had only one item at our disposal that measured general computer screen time without differentiation of online/offline use. However, prior research has clearly shown that computer use is very likely associated with internet use (Vaala & Bleakley, 2015). Nonetheless, we admit this as a limit that is mentioned in the discussion in the following way:

In addition, digital screen time did not measure internet use per se. Nonetheless, prior research as clearly showed that digital screen time predominantly overlap with time spent online (Vaala& Bleakley, 2015).

Reviewer 2 Report

The work called „The associations between family-related factors and excessive internet use in adolescents“ presents interesting insights from a large scale study conducted in Slovakia. Although I think that the findings add something of relevance to the literature, I have a number of amendments. Among others, much more literature needs to be covered (theory!). Of note: The literature put forward by myself in this review function just as examples and the authors can decide for themselves to include it or not. They also can decide to cite other relevant work from the literature instead.

Line 30/31: After this sentence a blank is missing. Moreover, you could add recent research dealing with this WHO diagnosis: This facilitated the inclusion of Internet Gaming Disorder in the ICD-11 (WHO).

Pontes, H. M., Schivinski, B., Sindermann, C., Li, M., Becker, B., Zhou, M., & Montag, C. (2019). Measurement and conceptualization of Gaming Disorder according to the World Health Organization framework: The development of the Gaming Disorder Test. International Journal of Mental Health and Addiction, 1-21.

Line 31: What do the authors mean with “Generalized internet addiction”? I highly recommend to strongly establish a theoretical background for the present research piece. As a starter the classic work by Davis (2001) helps. Then the perhaps best model at the moment - the I-PACE model - could be added, together with empirical evidence.

Brand, M., Young, K. S., Laier, C., Wölfling, K., & Potenza, M. N. (2016). Integrating psychological and neurobiological considerations regarding the development and maintenance of specific Internet-use disorders: An Interaction of Person-Affect-Cognition-Execution (I-PACE) model. Neuroscience & Biobehavioral Reviews71, 252-266.

Brand, M., Wegmann, E., Stark, R., Müller, A., Wölfling, K., Robbins, T. W., & Potenza, M. N. (2019). The Interaction of Person-Affect-Cognition-Execution (I-PACE) model for addictive behaviors: Update, generalization to addictive behaviors beyond Internet-use disorders, and specification of the process character of addictive behaviors. Neuroscience & Biobehavioral Reviews.

Davis, R. A. (2001). A cognitive-behavioral model of pathological Internet use. Computers in human behavior17(2), 187-195.

As an additional term (in line with the WHO diagnosis), the authors could also speak of Internet Use Disorder.

Line 47/48: As risk factors (low) self-directedness should be mentioned.

Choi, J. S., Park, S. M., Roh, M. S., Lee, J. Y., Park, C. B., Hwang, J. Y., ... & Jung, H. Y. (2014). Dysfunctional inhibitory control and impulsivity in Internet addiction. Psychiatry research215(2), 424-428.

Sariyska, R., Reuter, M., Bey, K., Sha, P., Li, M., Chen, Y. F., ... & Feldmann, M. (2014). Self-esteem, personality and internet addiction: a cross-cultural comparison study. Personality and Individual Differences61, 28-33.

The hypothesis mentioned in line 76 to 78 is only in parts backed up by the literature provided in the text. E.g. works on social support have not been good covered.

Hardie, E., & Tee, M. Y. (2007). Excessive Internet use: The role of personality, loneliness and social support networks in Internet Addiction. Australian Journal of Emerging Technologies & Society5(1).

Line 98: EUI was – it must be EIU

Line 105: I am not sure, but I think the “spend” should be “spent” here.

Line 107: preceding12 months. (a blank is missing)

Line 122-124. Several blanks are missing.

Methods: Is there an IRB approval for the present study? In particular this is relevant, because children/adolescents have been tested and this work touches a clinical phenomenon.

As this work has been carried out in Slovakia – what language versions of the questionnaires have been rolled out? Who did the translations? If the authors did the translations – how did they conduct the proper translation process to make sure that the content is appropriately measured?

Line 130/131: “The variable Parental Overprotection was constructed as a mean score of items for paternal overprotection and maternal overprotection.” Do I understand correctly that also scores for parental/maternal overprotection could be established separately? Then the question arises, if also in the analysis the separate scores should be considered?

I think it should be mentioned somewhere, that all measures were filled in by the children/adolescents only. Correct? Because in other works parents are asked about their parenting style, etc. …

Statistical analysis: The author mention that they present Spearman correlations, but use unpaired t-tests. They need to explain much more in detail what variables met requirements for parametric tests and if not: non-parametric testing needs to be conducted. Beyond this, the authors should in detail report which variables were significantly impacted by gender/age. Probably also interaction effects with gender appear and these need to be reported and tested.

Line 174-176: “An independent-sample t-test was conducted to compare the level of excessive internet use in boys (M = 1.62, SD = 0.64) and girls (M = 1.62, SD = 0.59). There was no significant difference (t(2645) = -0.18, p> 0.05, d = 0.01).” The authors write that EIU was highly skewed, then they need to do Mann-Whitney-U here.

In Table 3, it would be a nice add to get information about how much variance is explained by each variable included in the regression model. This provided in the text, but not the table.

I am not sure what regression model exactly has been computed. Hierarchical with enter methods? Please provide the reader with more information here.

210/211: “Second, our results show a mix of similarities and dissimilarities between EIU and other adolescent problematic behaviors, like drinking and delinquency.” This statement might only in part be valid, because it is heavily debated how much EIU and others additive tendencies overlap. It might also help to review some work on EIU and drinking/smoking, if the authors want to make an argument.

A further limitation is that the authors “only” investigated unspecified IUD. This needs to be more strongly discussed (see Davis’ model and the I-PACE model again).

Author Response

The work called „The associations between family-related factors and excessive internet use in adolescents“ presents interesting insights from a large scale study conducted in Slovakia. Although I think that the findings add something of relevance to the literature, I have a number of amendments. Among others, much more literature needs to be covered (theory!). Of note: The literature put forward by myself in this review function just as examples and the authors can decide for themselves to include it or not. They also can decide to cite other relevant work from the literature instead.

RESPONSE:

First of all, thank you very much for your supportive, insightful, and constructive feedback. Many of your comments were useful and to the point. In response to all of them, we have revised our manuscript and consider it much improved as a result.

We thank for the commentary a literature suggestion. We incorporated 12 new articles into the text, some of them those suggested.

Line 30/31: After this sentence a blank is missing. Moreover, you could add recent research dealing with this WHO diagnosis: This facilitated the inclusion of Internet Gaming Disorder in the ICD-11 (WHO).

Pontes, H. M., Schivinski, B., Sindermann, C., Li, M., Becker, B., Zhou, M., & Montag, C. (2019). Measurement and conceptualization of Gaming Disorder according to the World Health Organization framework: The development of the Gaming Disorder Test. International Journal of Mental Health and Addiction, 1-21.

Line 31: What do the authors mean with “Generalized internet addiction”? I highly recommend to strongly establish a theoretical background for the present research piece. As a starter the classic work by Davis (2001) helps. Then the perhaps best model at the moment - the I-PACE model - could be added, together with empirical evidence.

RESPONSE:

We have included these theoretical papers and addressed this issue in several parts of the introduction as well as in the discussion. We have also tried to clarify the original text as we acknowledge it was not well explained.  The paragraph about definition of EIU now states:

In our study, EIU represents an umbrella term for any excessive online behaviour. Although some older literature suggests the existence of generalised internet addictions [2], more current literature leans towards specific internet use disorders [4]. This has been supported by empirical findings that generalised and specific internet addiction mostly overlap [5], and that some online applications, especially online gaming [6] and social networking [7] contribute to this phenomenon more than others.

Brand, M., Young, K. S., Laier, C., Wölfling, K., & Potenza, M. N. (2016). Integrating psychological and neurobiological considerations regarding the development and maintenance of specific Internet-use disorders: An Interaction of Person-Affect-Cognition-Execution (I-PACE) model. Neuroscience &Biobehavioral Reviews71, 252-266.

Brand, M., Wegmann, E., Stark, R., Müller, A., Wölfling, K., Robbins, T. W., & Potenza, M. N. (2019). The Interaction of Person-Affect-Cognition-Execution (I-PACE) model for addictive behaviors: Update, generalization to addictive behaviors beyond Internet-use disorders, and specification of the process character of addictive behaviors. Neuroscience &Biobehavioral Reviews.

Davis, R. A. (2001). A cognitive-behavioral model of pathological Internet use. Computers in human behavior17(2), 187-195.

RESPONSE:

These theories were incorporated in both introduction:

Prior research has mostly investigated various individual risk factors that range from low self-esteem and self-efficacy [18,19], self-directedness [5]), neuroticism [20], and loneliness [21] to attention difficulties [22]. As such, the role of social environment is neglected in models that describing both generalised and specific disordered internet use –only the role of social cognition (e.g., feelings of isolation, perceived lack of social support) is mentioned in the I-PACE model [4] and the Cognitive-Behavioral Model of Pathological Internet Use [2]. However, theories of problematic adolescent behaviour, e.g. the Problem Behavior Theory [23] and theories for media effects, e.g. the Differential Susceptibility to Media Effects Model [24] propose stronger consideration of the environmental influence, notably the family, which may both predictand moderate adolescent behaviour, including patterns of internet use.

And in the discussion:

It is likely that the strongest influences for EIU are personality and psychological factors, like ADHD, lower self-esteem, and emotional difficulties in general [41], which is also suggested by theoretical models like the I-PACE [4],and cognitive-behavioural models [2,57]. Since Environmental factors play a non-negligible role in adolescent lives and they should be incorporated into theoretical models of EIU and specific internet disorders as  has been done in general media-effect models ( e.g., Differential Susceptibility of Media-Effects Model[24])

As an additional term (in line with the WHO diagnosis), the authors could also speak of Internet Use Disorder.

RESPONSE:

This term has been included together with appropriate citation:

Generalized internet addiction has not reached such a consensus in the scientific community and various terms have been proposed (e.g. internet addiction, compulsive internet use [1]; pathological internet use [2]; problematic internet use [3]; internet use disorder (Brand et al., 2016).

Line 47/48: As risk factors (low) self-directedness should be mentioned.

RESPONSE:

This term has been included together with appropriate citation

Prior research has mostly investigated various individual risk factors that range from low self-esteem and self-efficacy [16,17], self-directedness (Montag, 2010) neuroticism [18], and loneliness [19] to attention difficulties [20].

Choi, J. S., Park, S. M., Roh, M. S., Lee, J. Y., Park, C. B., Hwang, J. Y., ... & Jung, H. Y. (2014). Dysfunctional inhibitory control and impulsivity in Internet addiction. Psychiatry research215(2), 424-428.

Sariyska, R., Reuter, M., Bey, K., Sha, P., Li, M., Chen, Y. F., ... & Feldmann, M. (2014). Self-esteem, personality and internet addiction: a cross-cultural comparison study. Personality and Individual Differences61, 28-33.

The hypothesis mentioned in line 76 to 78 is only in parts backed up by the literature provided in the text. E.g. works on social support have not been good covered.

RESPONSE:

We have included the role of social support into the introduction together with appropriate citation:

Parents face challenges as their adolescent grows. Adolescents develop their own individuality and learn to take responsibility for their own actions. Their needs for privacy and independence gain importance [33], but at the same time compete with needs for closeness and relatedness, which still remains significant [34]. Insufficient social support in this developmental stage can intensify feelings of loneliness or social isolation, which often appear in adolescence [35]. Indeed, the levels of social support have been found to be associated with EIU [36].

Hardie, E., & Tee, M. Y. (2007). Excessive Internet use: The role of personality, loneliness and social support networks in Internet Addiction. Australian Journal of Emerging Technologies & Society5(1).

Line 98: EUI was – it must be EIU

 RESPONSE:

This has been fixed.

Line 105: I am not sure, but I think the “spend” should be “spent” here.

  RESPONSE:

This has been fixed.

Line 107: preceding12 months. (a blank is missing)

 RESPONSE:

This has been fixed.

Line 122-124. Several blanks are missing.

RESPONSE:

all typo has been corrected. However, the missing blanks is an issue connected to the use of IJERPH template and missing blanks in various parts of the text are created whenever the file is open. We will assure it is completely solved before publication.

Methods: Is there an IRB approval for the present study? In particular this is relevant, because children/adolescents have been tested and this work touches a clinical phenomenon.

RESPONSE:

We have such approval (the study was approved by the Ethics Committee of the Medical Faculty at the P.J. Safarik University in Kosice) and have included it into the text:

The study was approved by the Ethics Committee of the Medical Faculty at the P.J. Safarik University in Kosice (No: 9/2012). All participation was fully voluntary and anonymous, parental consent was obtained before the administration.

As this work has been carried out in Slovakia – what language versions of the questionnaires have been rolled out? Who did the translations? If the authors did the translations – how did they conduct the proper translation process to make sure that the content is appropriately measured?

RESPONSE:

Measurement section now includes description of the procedures – all scales were translated from English to Slovak and back translated. Special committee was established for checking the accuracy. The procedure is unified within HBSC network:

HBSC followed strict methodology of the survey which also applied to the translation process. All measures used in the study were translated from English to national Slovak language using the back-translation procedure. Moreover, back-translations are also independently checked by a translation team within HBSC network specifically established for this task.

Line 130/131: “The variable Parental Overprotection was constructed as a mean score of items for paternal overprotection and maternal overprotection.” Do I understand correctly that also scores for parental/maternal overprotection could be established separately? Then the question arises, if also in the analysis the separate scores should be considered?

RESPONSE:

Yes, it is possible, and we originally considered this solution and before the first submission of this manuscript, we conducted analyse separately for parental/maternal overprotection. We found that directions of the associations between overprotection and EIU were similar for both mothers and fathers. However, statistical power was lower so that we decided to keep this independent variable as a joint variable for both the parents. In addition, this combined parental variable allowed us to include into analyses also those respondents who had only one parent.

I think it should be mentioned somewhere, that all measures were filled in by the children/adolescents only. Correct? Because in other works parents are asked about their parenting style, etc. …

RESPONSE:

Yes, only adolescent filled in the questionnaire. We included statements:

A total of 10 179 adolescents aged 11-15 participated in the survey and filled the questionnaire. Measures listed below were offered on to these children.

Statistical analysis: The author mention that they present Spearman correlations, but use unpaired t-tests. They need to explain much more in detail what variables met requirements for parametric tests and if not: non-parametric testing needs to be conducted. Beyond this, the authors should in detail report which variables were significantly impacted by gender/age. Probably also interaction effects with gender appear and these need to be reported and tested.

RESPONSE:

Non-parametric tests were used only in case of correlations and Mann-Whitney U tests. We employed the regression since the criterion for normal distribution of residuals was fulfilled. Therefore, we did not have to use its non-parametric variation.  Nonetheless, we included this information:

In case of sufficiently large sample size regression analysis provides valid results even for non-normally distributed variables (Lumley, Diehr, Emerson, & Chen, 2002).

Line 174-176: “An independent-sample t-test was conducted to compare the level of excessive internet use in boys (M = 1.62, SD = 0.64) and girls (M = 1.62, SD = 0.59). There was no significant difference (t(2645) = -0.18, p> 0.05, d = 0.01).” The authors write that EIU was highly skewed, then they need to do Mann-Whitney-U here.

 RESPONSE:

Thank you for this point. We included Mann-Whitney-U tests. We also included information about skewness and kurtosis into Table 1.

In Table 3, it would be a nice add to get information about how much variance is explained by each variable included in the regression model. This provided in the text, but not the table.

RESPONSE:

The information was already available in Table 3 as an adjusted R square.

I am not sure what regression model exactly has been computed. Hierarchical with enter methods? Please provide the reader with more information here.

RESPONSE:

Yes. We have included the information that the enter method was used:

To determine the associations between excessive internet use and family factors, we conducted a hierarchical multiple regression that controlled for the gender and age of the participants and for the frequency of computer use. Variables were entered into the regression in two steps using the Enter method.

210/211: “Second, our results show a mix of similarities and dissimilarities between EIU and other adolescent problematic behaviors, like drinking and delinquency.” This statement might only in part be valid, because it is heavily debated how much EIU and others additive tendencies overlap. It might also help to review some work on EIU and drinking/smoking, if the authors want to make an argument.

RESPONSE:

We extended the discussion part by incorporating more information about the relationship between adolescent problematic behaviour and EIU. We wanted to add to the discussion on similarities and dissimilarities between EIU and other problematic behaviours in youth by pointing out that our findings on EIU showed that EIU constitutes a slightly different form of problematic behaviour since it was associated with different parenting strategies. We also noted that, in our opinion, the theoretical models of internet-based disorders do not sufficiently reflect the family/environmental influence of adolescent problematic behaviour.  

The largely reworked paragraph is now as follows:

Second, from the perspective of associations among family environmental factors and problematic behaviour in youth, our results show a mix of similarities and dissimilarities between EIU and other traditional problematic adolescent behaviours. In the case of traditional antisocial adolescent behaviours, such as drinking alcohol and delinquency, parental monitoring and care have generally been identified as protective factors, while parenting styles that harm the confidence and independence of children were found to be risk factors[50,51]. On the other hand, antisocial behaviours were often explained decreased family monitoring and involvement [52,51]. In this respect, EIU seems to be distinct from traditional antisocial and problematic behaviours – our results showed that overprotection and increased time at home and in the family environment (in other words, decreased time in peer groups) increase EIU. The distinction between EIU and problematic adolescent behaviour is further supported by De Looze and his colleagues [53], who found that the overall trend in the decrease of substance use in recent adolescents is not related to increased use of digital media; thus, EIU is not simply a replacement for other forms of problematic behaviour. Despite the fact that some studies found an association between EIU and various conduct problems [54,55],and that problematic adolescent behaviours and EIU share some underpinnings with ADHD [56], our study points to the need for specific parenting approaches whenaddressing EIU and when addressing traditional problematic behaviours. The mutual relatedness of EIU and other forms of problematic behaviours should be the subject of a more thorough investigation.

A further limitation is that the authors “only” investigated unspecified IUD. This needs to be more strongly discussed (see Davis’ model and the I-PACE model again).

RESPONSE:

We have addressed this issue as a limit of this study:

And, lastly, the main variable, EIU, although used in many previous studies and validated in most European countries, consists of only five items and does not allow for an in-depth investigation of the excessive use of various applications. This may be problematic from the perspective of the ongoing discussion on generalised vs specific internet-based disorders. In this respect, our study lacked to differentiate which applications contribute to EIU.

Round 2

Reviewer 2 Report

I think the authors did a good job in answering my amdendments.